# Identifying Preoperative Clinical Characteristics of Unexpected Gastrointestinal Perforation in Infants—A Retrospective Cohort Study

**DOI:** 10.3390/children11050505

**Published:** 2024-04-23

**Authors:** Adinda G. H. Pijpers, Ramon R. Gorter, Laurens D. Eeftinck Schattenkerk, Joost van Schuppen, Chris H. P. van den Akker, Sylvie Vanhamel, Ernest L. W. van Heurn, Gijsbert D. Musters, Joep P. M. Derikx

**Affiliations:** 1Department of Pediatric Surgery, Emma Children’s Hospital, Amsterdam UMC, University of Amsterdam, Meibergdreef 9, 1105 AZ Amsterdam, The Netherlands; 2Amsterdam Gastroenterology Endocrinology Metabolism Research Institute, 1105 AZ Amsterdam, The Netherlands; 3Amsterdam Reproduction & Development Research Institute, 1105 AZ Amsterdam, The Netherlands; 4Department of Radiology and Nuclear Medicine, Amsterdam UMC, University of Amsterdam, Meibergdreef 9, 1105 AZ Amsterdam, The Netherlands; 5Department of Neonatology, Emma Children’s Hospital, Amsterdam UMC, University of Amsterdam, Meibergdreef 9, 1105 AZ Amsterdam, The Netherlands; 6Department of Surgery, Zaans Medisch Centrum, Koningin Julianaplein 58, 1502 DV Zaandam, The Netherlands

**Keywords:** pneumoperitoneum, perforation, infants

## Abstract

Background: Infants presenting with unexpected pneumoperitoneum upon abdominal X-ray, indicating a gastrointestinal perforation (GIP), have a surgical emergency with potential morbidity and mortality. Preoperative determination of the location of perforation is challenging but will aid the surgeon in optimizing the surgical strategy, as colon perforations are more challenging than small bowel perforations. Therefore, the aim of this study is to provide an overview of preoperative patient characteristics, determine the differences between the small bowel and colon, and determine underlying causes in a cohort of infants with unexpected GIP. Methods: All infants (age ≤ 6 months) who presented at our center with unexpected pneumoperitoneum (no signs of pneumatosis before) undergoing surgery between 1996 and 2024 were retrospectively included. The differences between the location of perforation were analyzed using chi-squared and t-tests. Bonferroni correction was used to adjust for multiple tests. Results: In total, 51 infants presented with unexpected pneumoperitoneum at our center, predominantly male (N = 36/51) and premature (N = 40/51). Among them, twenty-six had small bowel, twenty-two colon, and three stomach perforations. Prematurity (*p* = 0.001), birthweight < 1000 g (*p* = 0.001), respiratory support (*p* = 0.001), and lower median arterial pH levels (*p* = 0.001) were more present in patients with small bowel perforation compared with colon perforations. Pneumatosis intestinalis was more present in patients with colon perforation (*p* = 0.004). All patients with Hirschsprung disease and cystic fibrosis had colon perforation. The final diagnoses were mainly focal intestinal perforations (N = 27/51) and necrotizing enterocolitis (N = 9/51). Conclusions: Infants with unexpected GIP, birthweight < 1000 g, and prematurity have more risk for small bowel perforation. In case of colon perforation, additional screening (for Hirschsprung and cystic fibrosis) should be considered.

## 1. Introduction

An infant presenting with pneumoperitoneum upon abdominal X-ray, suggesting a gastrointestinal perforation (GIP), is a surgical emergency that can lead to significant morbidity and mortality, especially in preterm infants [1,2,3,4,5,6]. In infants, GIP may be caused by a variety of factors, including necrotizing enterocolitis (NEC), focal intestinal perforation (FIP), ingestion of a foreign object, and blow-out due to atresia or Hirschsprung disease [3,7,8,9]. In some of these factors, such as necrotizing enterocolitis, the patient is already diagnosed based on diagnostic findings, like pneumatosis intestinalis upon abdominal X-ray. The cause of the GIP is then obviously linked to the necrotizing enterocolitis. However, there is a minority of patients who have no clinical symptoms at the time of the presence of pneumoperitoneum upon abdominal X-ray. In these patients, this unexpected GIP is caught by surprise, and an emergency surgery is conducted without knowing the underlying condition.

The diagnosis of unexpected GIP is typically made using a conventional abdominal X-ray or ultrasound. However, identification of the location of the unexpected GIP and thus differentiating between small bowel and colon perforation can be difficult, as the symptoms and signs may be similar, and radiological imaging alone may not be sufficient for accurate differentiation [1]. Distinguishing between these two locations of perforations is important as they can have different clinical presentations, management approaches, and prognoses. The surgical approach to the small bowel is relatively quick and the bowel is easily visible. However, it takes longer to gently prepare the colon because it is more fixated, especially in the flexures of the colon. This is one reason why preoperative estimation of the location of the unexpected perforation could help the decision-making of the surgeon and counseling of the parents. Furthermore, the cause of a small bowel perforation might be different from colon perforations. For example, a colon perforation (due to rectal irrigation) can occur in Hirschsprung disease or blow-out of the terminal ileum in the case of meconium ileus [10]. In order to anticipate which additional diagnostic interventions should be used perioperatively (for instance biopsies), early differentiation between small bowel and colon perforation might be helpful. For example, in case of a colon perforation, full-thickness biopsy could be taken intra-operatively in order to diagnose Hirschsprung disease or additional genetic testing for cystic fibrosis in case of meconium ileus.

Therefore, the aim of this study is to provide an overview of the preoperative clinical characteristics of infants with an unexpected GIP. Second, we aim to compare preoperative differences between unexpected small bowel and colon perforations in order to early differentiate between the two locations of GIP. Third, we investigated potential underlying causes in these patients with unexpected GIP in order to guide which diagnostic interventions should be implemented during primary surgery for GIP.

## 2. Materials and Methods

All consecutive infants less than six months old presenting with an unexpected pneumoperitoneum confirmed by radiological imaging who underwent surgery between July 1996 and March 2024 were retrospectively identified. Infants with a pre-existing underlying diagnosis that could result in a pneumoperitoneum, such as necrotizing enterocolitis (NEC), were excluded from this study since, in these infants, the GIP was most often not unexpected. The patient was also excluded if no surgical or radiological reports were available. This study was reviewed and approved by the medical ethical committee (reference: W18_233#18.278, July 2018). Parents were sent an opt-out letter for consent, and following consent, patient records were checked for eligibility (SV, AP both MDs). In case of doubts or inconsistency, a pediatric surgeon (JD) was consulted for final judgment. All abdominal radiographs and ultrasounds were blindly reviewed and analyzed by a pediatric radiologist (JvS) and pediatric surgeon (JD).

### 2.1. Data Extraction

Patient records were searched for the description of gender, gestational age in weeks, prematurity (defined as gestational age less than 37 weeks), age at surgery in days, birthweight (stratified in <1000 g, 1001–1500 g, and >1500 g), associated anomalies, length of hospital stay in days, and mortality within 30 days after surgery. A pneumoperitoneum was defined as the presence of air in the peritoneal cavity diagnosed by a pediatric radiologist using X-ray or ultrasound [11,12]. The abdominal X-rays and ultrasounds were retrospectively scored on the amount of free abdominal air (massive, locally, minimal), presence of dilated bowel (present or not present), and pneumatosis intestinalis (present or not present). When images were not available, radiological reports were scored. Surgical reports were searched for the location of the perforation, and the underlying causes of the perforation were extracted. The final diagnosis was based on a combination of clinical, radiological, and pathological factors. Information concerning associated anomalies was divided into organ systems, including cardiovascular; vertebral; renal; genito–urinary tract; eye, ear, and neck; esophageal; gastrointestinal; limbs; cleft, lip, and palate; nervous system; and genetics. In order to compare the preoperative characteristics, the cohort was divided into two groups; small bowel perforations and colon perforations.

The presence of sepsis, shock, and SIRS (Systematic Inflammatory Response Syndrome) were scored according to the Pediatric Sepsis Consensus (PDC) criteria, as reference values for vital parameters and laboratory findings were collected from the patient records [13]. Sepsis was scored when this was culture-proven. Vital parameters were collected in the 24 h before surgery and were corrected for the (gestational) age of the child [13].

### 2.2. Outcome Parameters

The primary outcome was to provide an overview of the characteristics of patients presenting with an unexpected pneumoperitoneum. Our secondary outcome was to compare these patient characteristics between small bowel and colon perforations. The third outcome was to investigate the underlying causes in these patients with unexpected GIP in order to know which diagnostics to use.

### 2.3. Statistical Analysis

Normally distributed data were reported as mean and standard deviation (SD), while non-normally distributed variables were reported as median with interquartile range (IQR). The distribution of the data was assessed using visual inspection through histogram analysis. The chi-squared test was used to test the significance of differences between perforations of the small bowel or colon, and if the assumptions of this test were not met, Fisher’s Exact test was used. For continuous data, Student’s *t*-test was used for normally distributed data, while non-normally distributed variables were tested with the Mann–Whitney U test. To correct for type-one error rate (false/positive correlations), Bonferroni correction was used to adjust for multiple comparisons. Statistical significance was defined as *p*-values of less than or equal to 0.05.

## 3. Results

A total of fifty-one infants presented with a pneumoperitoneum upon abdominal X-ray, suggesting an unexpected GIP, of which twenty-six were already admitted to our university hospital during the discovery of the pneumoperitoneum, twenty-three were referred to our university hospital, and two presented at the emergency department.

Of the 51 infants, 40 (78.4%) were born prematurely, 20 (39.2%) had a birth weight of less than 1000 g, and 34 (70.8%) were male. Twin births were seen in 15 of the 51 (29.5%) patients. The median postnatal age at surgery was five days (IQR: 3–10). Enteral feeding was already started in 48 of the 51 (94.1%) patients. The first meconium passage was delayed by more than 48 h after birth in twenty of the fifty-one patients (39.2%), and in four patients (7.8%), bloody stool was discovered. Preoperatively, eight of the fifty-one patients (15.7%, missing: nine) presented with sepsis (all-culture-proven sepsis). Preoperative respiratory support was given to 40 of the 51 patients (78.4%), including 17 (42.5%) who received non-invasive ventilation and 21 who received mechanical ventilation (52.5%). In 14 patients (35.0%), both methods of respiratory support were applied.

Vital parameters were measured within 24 h preoperatively and showed abnormal body temperature in five of the fifty-one patients (9.8%), of which three patients had hyperthermia and two had hypothermia. Short respiratory arrests were present in eighteen patients (35.3%), and seven had tachypnea (13.7%). Hypertension was found in three patients (5.8%), and hypotension in eight (15.7%). An abnormal heart rate was found in twelve patients (23.5%), of which ten had tachycardia (19.6%), and two had bradycardia (3.9%). Blood results were tested within 24 h before surgery, showing median values of arterial pH 7.30 (IQR: 7.19–7.36), lactate 2.6 mmol/L (IQR: 1.7–3.2), CRP (C-reactive protein) 20.0 mg/L (IQR: 5.9–33.0), leucocytes 9.5 10E9/L (IQR: 5.9–25.1), platelets 169 10E9/L (IQR: 87–235), and Hb 8.4 mmol/L (IQR: 7.3–10.0). The rest of the baseline characteristics are shown in Table 1.

### 3.1. Radiological Findings

The diagnosis of unexpected GIP was made using abdominal X-rays in 45 of the 51 patients (88.2%). Five of the fifty-one (9.8%) were diagnosed using abdominal ultrasound and one during a fistulogram (2.0%). In four patients, the radiological report was only available without the presence of the radiological image. Radiological imaging showed a massive amount of abdominal free air in 35 of 47 patients (88.2%), and multiple dilated bowel loops in 31 of 47 patients (66.0%). Pneumatosis intestinalis was also present upon abdominal X-ray or ultrasound at the time of diagnosing pneumoperitoneum in 10 patients (21.3%) and not seen on radiological imaging before. Ascites was found in 16 of 47 patients (34.0%). Radiological variables are shown in Table 2. In Figure 1A,B, massive abdominal free air in an infant is shown. Free abdominal air in front of the liver upon abdominal ultrasound is shown in Figure 1C.

### 3.2. Intraoperative Findings and Postoperative Diagnosis

The unexpected intestinal perforations were located in the small bowel in 26 of the 51 patients (50.9%) and in the colon in 22 of the 51 (43.1%) patients. The remaining three patients had a stomach perforation. Multiple perforations (varying between two and four) were seen in nine of the fifty-one (17.3%) patients. We did not encounter combinations of perforations in both the small bowel and colon. In the majority of the patients, a stoma was created (31.4%, from which two patients received an abdominal drain for peritoneal before stoma creation), followed by primary anastomosis (25.5%) or a combination of both (27.5%). In eight patients, the bowel was primarily closed with stitches. In five out of fifty-one (9.8%) patients, mapping (multiple full-thickness biopsies taken on different locations of the colon) was performed during the primary surgery. And in 25 of the 51 patients (49.0%), a trans-anal full-thickness biopsy was taken. In 33 of the 51 patients, additional tests were performed, including pathological evaluation of the resected tissue (47.0%), cystic fibrosis diagnostics (9.8%), rectal suction biopsy (2.0%), and clinical genetics consultation (5.9%). The final diagnosis was focal intestinal perforation in twenty-seven of the fifty-one patients (52.9%), followed by NEC in nine out of the fifty-one patients (17.6%). The remaining diagnoses were meconium ileus in five out of fifty-one patients, of which one had cystic fibrosis, there was iatrogenic injury in four patients (one caused by gastric perforation of a nasogastric tube, one caused by mechanical ventilation leading to stomach perforation, one traumatic birth causing a small bowel perforation, and one colon perforation caused by rectal irrigation), Hirschsprung disease in four patients, and milk-curd syndrome and appendicitis occurred in one patient. Associated anomalies were seen in ten of the fifty-one patients (19.6%), with major cardiac anomalies requiring surgery in four of the patients (7.8%). The intraoperative findings and final diagnosis are shown in Table 3.

### 3.3. Small Bowel versus Colon Perforations

Table 4 provides an overview of the location of unexpected perforations divided into two groups: small bowel (N = 26) perforations and colon (N = 22) perforations. In the small bowel perforations group, there were significantly more prematurely born patients (*p* = 0.001) as well as significantly more patients with a birthweight < 1000 g (*p* = 0.001) compared to the colon perforations. Associated anomalies occurred more in patients with a colon perforation but this was not significant after Bonferroni correction (*p* = 0.022). There were no significant differences between the two groups for male gender (*p* = 0.202), mode of delivery (*p* = 0.041), sepsis (*p* = 0.255), major cardiac anomaly (*p* = 0.221), age at surgery (*p* = 0.164), and death within 30 days (*p* = 0.509). Median arterial pH levels were lower in the group of patients with small bowel perforations (7.20) compared to the group with colon perforations (7.36). Other blood results did not significantly differ between the two groups. Radiological characteristics showed that pneumatosis intestinalis was significantly more often seen upon abdominal X-ray or ultrasound in patients with colon perforations compared to small bowel perforations (*p* = 0.004). There were no significant differences in the amount of abdominal free air, the amount of dilated bowel loops, and the presence of ascites between the two groups. The final diagnosis showed a difference in the group of Hirschsprung patients. Hirschsprung was more present in patients with colon perforation.

## 4. Discussion

In our cohort of 51 patients who presented with unexpected GIP, we identified twenty-six small bowel perforations (50.9%), twenty-two colon perforations (43.1%), and three stomach perforations (6.0%). The final diagnoses mainly included FIP (52.9%) and NEC (17.6%). We compared two primary locations of unexpected perforations—small bowel and colon perforations—and found that premature birth, birthweight under 1000 g, the use of respiratory support, and lower median arterial pH concentrations were significantly more present in patients who had small bowel perforations rather than colon perforations. In contrast, pneumatosis intestinalis was significantly more present in patients with colon perforations. All patients with Hirschsprung disease presented with a colon perforation.

According to our cohort, preterm-born infants or with birthweights under 1000 g present more often had unexpected small bowel perforations than colon perforations. This is in line with previous studies [14,15,16,17,18]. Of course, NEC and FIP are conditions associated with preterm birth and an extremely low birth weight [14]. This is attributable to the underlying pathology that more premature infants with a lower birth weight have thinner intestinal walls. These underdeveloped intestinal muscular walls of the small bowel are dependent on insulin-like growth factor-I (IFG-I) [19]. It has been established that IGF-I is the main growth factor in charge of submucosal thickness and tensile strength of the intestinal wall [14,20]. IGF-I is tightly regulated by a number of factors throughout fetal development, including hormones, growth factors, and nutrition [14,19]. The two factors that are most closely connected with intestinal wall thinness due to IGF-I paucity are premature birth and low birth weight and are therefore the best predictors of FIP [21,22].

The use of noninvasive mechanical ventilation (NIMV) strategies has reduced the need for invasive mechanical ventilation in recent years [17]. However, the use of NIMV has some drawbacks. It has been suggested that the use of high positive inspiratory pressure (PIP) may increase the risk of perforation in the stomach and proximal intestinal tract due to the high pressure generated by the ventilator on the intestinal wall [7]. In our cohort, we observed three cases of unexpected stomach perforations, two of which had received both invasive and noninvasive mechanical ventilation. Furthermore, respiratory support was more common among infants with small bowel perforations compared to those with colon perforations. Noninvasive mechanical ventilation with high PIP may cause elevated pressure in the upper digestive tract, which could lead to perforation [7]. However, the association between NIMV and small bowel perforations may also be explained by the fact that premature birth, which is a risk factor for small bowel perforation, often requires respiratory support due to associated respiratory issues.

We found that patients with the presence of pneumatosis intestinalis upon abdominal X-ray or ultrasound have significantly more colon perforations compared to small bowel perforations. The pathophysiology of pneumatosis intestinalis is based on bacterial translocation through the mucosa and the formation of hydrogen gas in the bowel wall [23]. To our knowledge, this is the first study that describes that pneumatosis intestinalis is more present in patients with colon perforations. Previous studies did show that pneumatosis intestinalis upon abdominal X-ray is more often seen in the colon [24]. It is therefore conceivable that the higher number of pneumatosis intestinalis in this group is due to the fact that pneumatosis is more easily visible when located in the colon. No hard conclusions can be drawn based on this small sample size. Future studies regarding the presence of pneumatosis intestinalis and the location of perforation should be indicated.

All previous statements taken into consideration, there are several preoperative characteristics that might help early differentiation between unexpected small bowel and colon perforations. In case of premature birth, birth weight under 1000 g, the use of respiratory support, and lower median arterial pH levels at presentation, small bowel perforation seems more likely. On the other hand, pneumatosis intestinalis was significantly more present in patients with colon perforations. Moreover, we found that associated anomalies are more present in patients with colon perforation. Cardiac anomalies (7.8%) and cystic fibrosis (2.0%) made up the majority of these anomalies. In addition, Hirschsprung disease is seen only in patients presenting with a colon perforation. It is crucial to identify these anomalies early so that appropriate additional diagnostics and treatment can be started. The gold standard in diagnostics of Hirschsprung disease is obtaining tissue for pathological examination. Tissue can easily be obtained during surgery with mapping of the colon or rectal biopsy [25,26,27]. It is preferable to perform diagnostics such as mapping or a biopsy promptly during the first procedure to avoid having to put a child under anesthesia again for further diagnostics after the operation. Preoperative patient characteristics like delayed meconium passage may be helpful. However, only fourteen out of the twenty-two patients in our cohort who had a colon perforation received a biopsy during primary surgery, resulting in seven patients who did not receive a biopsy. The early screening could be improved here to prevent additional postoperative biopsy. The screening of cystic fibrosis typically consists of a CFTR gene test [28]. This would be indicated when a meconium ileus is present, including thickened meconium present in the terminal ileum in combination with a microcolon [29,30]. The decision to perform screening for cystic fibrosis is also influenced by any extra symptoms, mainly of pulmonary origin. This demonstrates that screening for Hirschsprung disease and cystic fibrosis is not always performed based on preoperative patient characteristics in combination with patient-specific information that is present at surgery. This screening could be improved to prevent the patient from requiring another intervention for diagnosis. Making predictions about which child should be screened could be enhanced with a larger prospective cohort study.

Our study has certain limitations. First, our sample size was relatively small due to the relatively rare condition, which may have made this study underpowered to identify significant differences between the two groups. Furthermore, this prevented the evaluation of patients with gastric perforation. Second, this study was conducted at a single center, which may limit the external validity of our results. Third, our study was retrospective in nature, which may introduce bias or confounding variables that we were unable to control for. In order to correct for type-one error rate (false/positive correlations), Bonferroni correction was used to adjust for multiple comparisons. Future research should focus on the identification of risk factors for unexpected GIP in a cohort with a larger sample size.

## 5. Conclusions

In conclusion, unexpected small bowel perforations and colon perforations present with different clinical characteristics. Premature birth, birthweight < 1000 g, NIMV, and a decrease in arterial pH levels create the risk of developing a small bowel perforation. On the other hand, pneumatosis intestinalis was significantly more present in patients with colon perforations. These preoperative characteristics should be taken into account in order to provide early differentiation between the location of unexpected perforation and screening for associated anomalies, such as Hirschsprung disease and cystic fibrosis, and could start as soon as possible.

## Figures and Tables

**Figure 1 children-11-00505-f001:**
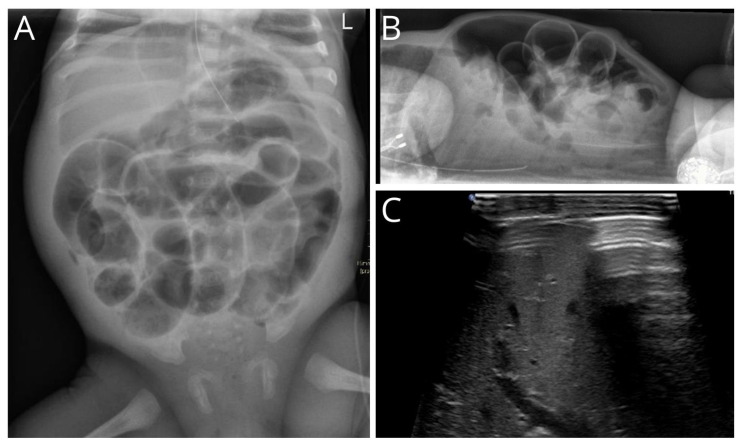
Abdominal X-ray and ultrasound of an infant with massive abdominal free air. (**A**) Anterior–posterior radiograph of the abdomen of an infant, including continuous diaphragm, football sign, Rigler sign, and suspicion of pneumatosis intestinalis in the right lower abdomen. (**B**) Lateral radiograph showing massive free air and tell-tail signs. (**C**) Reverbaration artifact of free air in front of liver.

**Table 1 children-11-00505-t001:** Baseline characteristics.

Variable	Total N = 51 N (%)
Presentation	
During hospital admittance	26 (51.0)
Referred to our center	23 (45.1)
Emergency department	2 (3.9)
Male	36 (70.6)
Premature	40 (78.4)
Median gestational age in weeks	28 (26.0–36.0)
Twins	15 (29.5)
Method of delivery	
Vaginal	33 (64.7)
C-section	18 (35.3)
Median age at surgery in days (IQR)	5 (3–10)
Birthweight groups	
<1000 g	20 (39.2)
1001–1500 g	10 (19.6)
>1500 g	21 (41.2)
Sepsis	8 (15.7)
Missing	11 (21.6)
Exteral feeding before perforation	48 (94.1)
Meconium > 48 h after birth	20 (39.2)
Abdominal distention	46 (90.2)
Bloody stool	4 (7.8)
Respiratory support	40 (78.4)
Mechanical ventilation	21 (52.5)
Non-invasive ventilation	17 (42.5)
Unclear	2 (5.0)
Respiratory rate	
Normal	17 (33.3)
Tachypnea	6 (11.7)
Increased incidences	19 (37.3)
Missing	9 (17.6)
Blood results, median	
Arterial pH (IQR)	7.30 (7.19–7.36)
Lactate, in mmol/L (IQR)	2.6 (1.7–3.2)
CRP, in mg/L (IQR)	20.0 (5.9–33.0)
Leucocytes, in 10E9/L (IQR)	9.5 (5.9–25.1)
Platelets, in 10E9/L (IQR)	169 (87–235)
Hb, in mmol/L (IQR)	8.4 (7.3–10.0)

N: Number, IQR: Interquartile range, CRP: C-reactive protein.

**Table 2 children-11-00505-t002:** Radiological characteristics.

Variable	Total N = 51 N (%)
Type of radiology	
X-ray	45 (88.2)
Ultrasound	5 (9.8)
Fistulogram	1 (2.0)
Median age at radiological imaging in days (IQR)	5 (3–9)
Amount of abdominal air	
Massive	35 (74.5)
Locally	9 (19.1)
Peel of air	3 (6.4)
Dilated bowel	
None	1 (2.1)
Some	15 (31.9)
Multiple	31 (66.0)
Pneumatosis intestinalis	10 (21.3)
Ascites	16 (34.0)

N: Number, IQR: Interquartile range.

**Table 3 children-11-00505-t003:** Intraoperative findings and final diagnosis.

Variable	Total N = 51 N (%)
Type of perforation	
Gastric	3 (6.0)
Small bowel	26 (50.9)
Colon	22 (43.1)
Multiple perforations	9 (17.3)
Resection of bowel	36 (70.6)
Intraoperative diagnostic	
Mapping	5 (9.8)
Trans-anal biopsy taken	25 (49.0)
Additional diagnostics	
Pathology of resected bowel	24 (47.0)
CF diagnostics ^1^	5 (9.8)
Rectal suction biopsy	1 (2.0)
Clinical genetics consultation	3 (5.9)
Mode of treatment	
Stoma creation ^2^	16 (31.4)
Primary anastomosis	13 (25.5)
Primary closure of bowel	8 (15.7)
Combination	14 (27.5)
Final diagnosis	
Focal intestinal perforation	27 (52.9)
Necrotizing enterocolitis	9 (17.6)
Meconium ileus	5 (9.8)
Iatrogenic injury	4 (7.8)
Hirschsprung disease	4 (7.8)
Milk-curd syndrome	1 (2.0)
Appendicitis	1 (2.0)
Associated anomaly	7 (13.7)
Cardiac	4 (7.8)
Cystic fibrosis	1 (2.0)
Trisomy 13	1 (2.0)
Congenital Disorders of Glycosylation	1 (2.0)

N: Number, CF: Cystic fibrosis. ^1^: N = 4 Cystic Fibrosis Transmembrane Conductance Regulator (CFTR) gen test, N = 1 Sweat test. ^2^: In two patients, an abdominal drain for peritoneal drainage was placed preoperatively.

**Table 4 children-11-00505-t004:** Small bowel versus colon perforations.

Variable	N = 26 Small Bowel Perforations	N = 22 Colon Perforations	*p*-Value
Male gender	17 (65.4)	18 (81.8)	0.202 ^1^
Prematurity	25 (96.2)	12 (54.5)	0.001 ^1^
Median gestational age	26.0 (25.0–27.3)	36.5 (31.8–39.0)	0.001
Median age at surgery	4.5 (3.0–7.0)	7.0 (2.8–25.3)	0.164
Birthweight groups			0.001 ^1^
<1000	15 (57.7)	4 (18.1)	
1000–1500	8 (30.8)	1 (4.5)	
>1500	3 (11.5)	17	
Mode of delivery			0.041 ^1^
Vaginal	14 (53.8)	18 (81.8)	
C-section	12 (46.2)	4 (18.1)	
Associated anomaly	1 (3.8)	6 (27.2)	0.022 ^1^
Respiratory support			0.001 ^1^
Mechanical ventilation	3 (11.5)	4 (18.1)	
Non-invasive ventilation	12 (46.2)	5 (22.7)	
Both	9 (34.6)	3 (13.6)	
Sepsis	3 (11.5)	5 (22.7)	0.255 ^1^
Major cardiac anomaly	1 (3.8)	3 (13.6)	0.221 ^1^
Mortality within 30 days	5 (19.2)	6 (27.2)	0.509 ^1^
Blood results, median			
Arterial pH (IQR)	7.20 (7.17–7.30)	7.36 (7.31–7.41)	0.001 ^2^
Lactate, in mmol/L (IQR)	2.2 (1.2–2.9)	2.8 (2.0–3.5)	0.109 ^2^
CRP, in mg/L (IQR)	8.0 (1.5–22.9)	30.5 (17.2–60.2)	0.012 ^2^
Leucocytes, in 10E9/L (IQR)	9.5 (7.3–37.0)	8.8 (5.0–15.5)	0.234 ^2^
Platelets, in 10E9/L (IQR)	155 (118–197)	137 (78–251)	0.863 ^2^
Hb, in mmol/L (IQR)	8.5 (7.3–9.3)	8.3 (7.2–10.5)	0.678 ^2^
Radiological characteristics			
Abdominal free air			0.009 ^1^
Massive	23 (88.5)	10 (45.5)	
Locally	2 (7.7)	6 (27.2)	
Minimally	0 (0.0)	3 (13.6)	
Dilated bowel			0.355 ^1^
Multiple	17 (65.4)	13 (59.1)	
Some	9 (34.6)	4 (18.1)	
None	0 (0.0)	1 (4.5)	
Pneumatosis intestinalis	2 (7.7)	8 (36.4)	0.004 ^1^
Ascites	7 (26.9)	7 (31.8)	0.402 ^1^
Final diagnosis			
Focal intestinal perforation	17 (65.4)	9 (40.9)	
Necrotizing enterocolitis	5 (19.2)	4 (18.1)	
Meconium ileus	3 (11.5)	2 (9.1)	
Iatrogenic injury	1 (3.8)	1(4.5)	
Hirschsprung disease	0 (0.0)	4 (18.1)	
Milk-curd syndrome	0 (0.0)	1 (4.5)	
Appendicitis	0 (0.0)	1 (4.5)	

N: Number, IQR: Interquartile range. ^1^: Chi-squared test, significance level after Bonferroni correction *p* = 0.004. ^2^: Mann–Whitney U test, significance level after Bonferroni correction *p* = 0.008.

## Data Availability

The data presented in this study are available upon request from the corresponding author. The data are not publicly available due to privacy.

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
