# Peer review of "Identifying Preoperative Clinical Characteristics of Unexpected Gastrointestinal Perforation in Infants—A Retrospective Cohort Study"

_children, 2024, doi:10.3390/children11050505_

Round 1
Reviewer 1 Report
Comments and Suggestions for Authors
In this article, Pijpers et al. conducted a single-center retrospective study to identify clinical characteristics associated with unexpected or spontaneous intestinal perforations. Furthermore, the authors sought to identify pre-operative characteristics that could help differentiate small bowel versus colon perforations, providing the rationale that such distinction could help with surgical planning as well as whether additional diagnostic evaluation is needed.
Overall, the article is well-written and presented logically. I have a few comments and suggestions.
-
An alternative to exploratory laparotomy for cases of unexpected gastrointestinal perforation in preterm infants, majority of which are focal intestinal perforations, is the placement of peritoneal drainage, a procedure that was not discussed at all in the paper. Is peritoneal drainage not performed at your local institution, or was cases that underwent peritoneal drainage not included in the study? I would suggest mentioning about peritoneal drainage and adding a paragraph in the discussion about speculation of how your findings could apply to other institutions where peritoneal drainage is an option.
-
Is sepsis culture-proven sepsis? I would specify if this were culture-proven or culture-negative sepsis.
-
Table 1. Median gestational age and median age at surgery in days had erroneously included the denominator of 48. Kindly correct.
-
Table 1. Median months follow-up: I would consider removing as it is not a pre-operative or baseline characteristic and does not add to the results.
-
Table 4. It would be more helpful to compare small intestinal vs colonic perforations by the median gestational age at birth, rather than the categorical variable of prematurity.
-
Table 4. Could you also compare the median age at surgery of the two groups? I am curious whether the small bowel perforations tend to occur earlier than the colonic perforations.
-
Table 4. Kindly provide number (percentages) for the data similar to that of the other tables.
Reviewer 2 Report
Comments and Suggestions for Authors
The authors provide an overview of preoperative patient characteristics, determine the differences between small and large bowel, and identify the underlying causes in a cohort of infants with unexpected gastrointestinal perforation.The study is interesting and may be of interest, but some objections need revision and clarification:
1. In the abstract, please state the objective without the subheadings (1), (2) and (3). Present it as one sentence.
2. The main shortcoming of this study is the relatively small sample size of infants included in the analysis. As this is a retrospective study, it is unclear why the infants from July 2022 to March 2024 were not included in the analysis to increase the sample size. I would advise extending the study period to March 2024.
3. Also, the authors should include a comparison group of healthy infants to compare the variables. In this way, this is a simple descriptive study.
4. To identify preoperative characteristics or risk factors for GIP, the authors are advized to perform additional statistical tests with logistic regression and multiple logistic regressions to confirm risk factors for GIP.
5. The authors have indicated two groups of patients in Table 4. These groups should be described in the methodology.
6. What statistical test was used to check the normality of the data distribution? I do not see any of the variables presented as mean (SD). Since it is clear from the methodology that not all variables are normally distributed, is it hard to believe that some of the variables are not normally distributed? Please clarify this!
7. The bibliography is incomplete. The authors should do a better literature search. More recent references should be cited. Minimum of 30 references the standard for an original article. Several important studies have been omitted and should be compared with obtained results and discussed: doi: 10.1007/s12262-016-1565-z., doi: 10.4240/wjgs.v9.i2.46..
8. It would be advizable if the authors have any X-rays or intraoperative photos to include some as support to the text in document.
Round 2
Reviewer 2 Report
Comments and Suggestions for Authors
I am satisfied with authors response and corrections made in manuscript. Manuscript is acceptable for publication as it stands.